# Envelope glycoprotein mobility on HIV-1 particles depends on the virus maturation state

Jakub Chojnacki [1], Dominic Waithe [2], Pablo Carravilla[3], Nerea Huarte[3], Silvia Galiani[1], Jörg Enderlein [4] & Christian Eggeling[1]

Human immunodeficiency virus type 1 (HIV-1) assembles as immature particles, which require the proteolytic cleavage of structural polyprotein Gag and the clustering of envelope glycoprotein Env for infectivity. The details of mechanisms underlying Env clustering remain unknown. Here, we determine molecular dynamics of Env on the surface of individual HIV-1 particles using scanning fluorescence correlation spectroscopy on a super-resolution STED microscope. We find that Env undergoes a maturation-induced increase in mobility, highlighting diffusion as one cause for Env clustering. This mobility increase is dependent on Gag-interacting Env tail but not on changes in viral envelope lipid order. Diffusion of Env and other envelope incorporated proteins in mature HIV-1 is two orders of magnitude slower than in the plasma membrane, indicating that HIV-1 envelope is intrinsically a low mobility environment, mainly due to its general high lipid order. Our results provide insights into dynamic properties of proteins on the surface of individual virus particles.

[1] MRC Human Immunology Unit, Weatherall Institute of Molecular Medicine, University of Oxford, OX3 9DS Oxford, UK. [2] Wolfson Imaging Centre, Weatherall Institute of Molecular Medicine, University of Oxford, OX3 9DS Oxford, UK. [3] Biofisika Institute (UPV/EHU, CSIC) and Department of Biochemistry and Molecular Biology, University of the Basque Country (UPV/EHU), P.O. Box 644, 48080 Bilbao, Spain. [4] Third Institute of Physics, Georg August University, 37077 Göttingen, Germany. Correspondence and requests for materials should be addressed to J.C. (email: jakub.chojnacki@rdm.ox.ac.uk) or to C.E. (email: christian.eggeling@rdm.ox.ac.uk)

During its assembly, human immunodeficiency virus type 1 (HIV-1) incorporates 7–10 copies of the trimeric viral envelope surface glycoprotein Env[1, 2] and ≈ 2400 copies of the main structural protein Gag[3]—a polyprotein that initially assembles into immature virus particles (Fig. 1a). Generation of the morphologically mature fully infectious virus particles requires a release of individual proteins (MA (matrix), CA (capsid), NC (nucleocapsid) and p6) from Gag in a series of tightly regulated cleavage steps catalysed by the viral protease (PR). These steps are thought to occur concurrently or shortly after virus particle budding from the plasma membrane of an infected cell, and the entire process, termed maturation, is a crucial step in the HIV-1 replication cycle and an important target in anti-HIV-1 therapy[4].

Despite possessing the same Env protein structure and count as fully mature virus, immature virus particles are non-infectious owing to their inability to enter the target cell[5, 6]. This effect appears to be caused by the stiffness of the immature Gag lattice, preventing membrane fusion[7] as well as maturation induced lateral reorganisation of Env[2]. In Env reorganisation study, PR-induced disassembly of Gag lattice was found to allow multi-clustered Env to coalesce into a single cluster in fully infectious mature particles. Such cluster formation may result from an increase in Env mobility upon maturation. This coupling of Env molecular rearrangements with virus maturation represents a novel opportunity to measure HIV-1 maturation kinetics. Although recent studies, utilising a novel photo-destructible protease inhibitor, were able to accurately estimate the overall HIV-1 maturation kinetics by immunoblot analysis of Gag[8] or by super-resolution STED microscopy imaging of Gag proteins in individual virus particles[9], Env mobility measurements may offer an alternative approach to estimate HIV-1 maturation kinetics in the context of the single virus without the need to incorporate tagged HIV-1 Gag molecules. Furthermore, intact HIV-1 virus particles represent a unique highly curved protein/lipid environment whose molecular dynamics have to date not been studied, mainly owing to their small size ( < 140 nm)[3, 10], which is below the resolution limit of conventional optical microscopy. Measurements of HIV-1 Env mobility would therefore enable for novel insights into the molecular dynamics of the virus envelope.

Super-resolution optical microscopy imaging has been used previously to reveal HIV-1 subviral structures[2, 9, 11]. However, measurements of the mobility of virus surface molecules require an observation technique with both a high spatial and temporal resolution as their diffusion coefficient may be comparable to plasma membrane proteins ( ≈ 0.1 μm²/s), and the spatial displacement is within the <140 nm diameter of the virus particle. STED microscopy achieves the required spatial resolution[12], but fails to record subsequent images with a frame rate fast enough to recover single-membrane protein dynamics. Studies of fast molecular dynamics on a STED microscope can be realized by combining it with single-molecule-based spectroscopic tools such as fluorescence correlation spectroscopy (FCS). In FCS, average diffusion coefficients are determined from the transit time of fluorescently tagged molecules through the observation spot[13]. STED-FCS enables the determination of molecular mobility for observation spot sizes below 60 nm in diameter[14], thus allowing for FCS measurements on individual virus particles. In a conventional single-point STED-FCS or even FCS measurements, which usually take a few seconds, the small pool of 7–10 tagged Env molecules on an individual virus would photobleach rapidly within less than a second, and therefore this measurement modality is unsuitable. On the other hand, scanning STED-FCS (sSTED-FCS) is an advanced implementation of STED-FCS where, unlike in single-point measurements, molecular transits are recorded by fast scanning of the STED microscope's

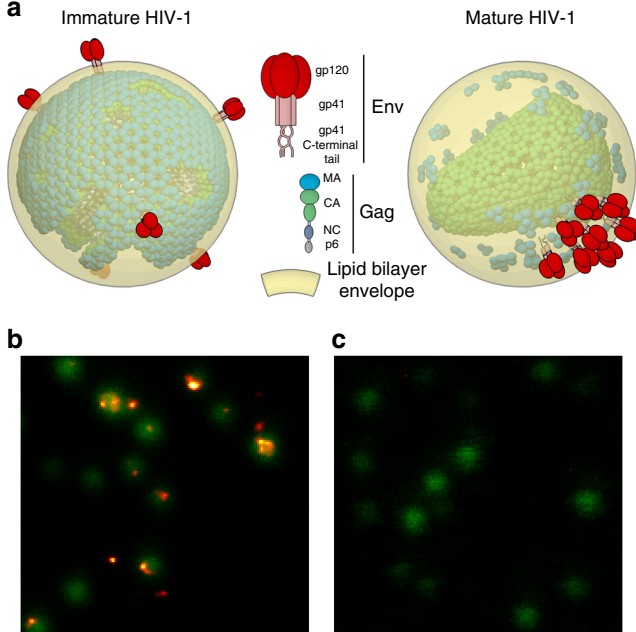

**Fig. 1** Env organisation in live unfixed HIV-1 particles. **a** Schematic illustration of Env and Gag distribution on mature and immature HIV-1. **b**, **c** Microscopy images of eGFP.Vpr (*green*, confocal) and Env (*orange*, STED) visualised by immunostaining in live unfixed mature wild-type **b** and Env(−) **c** HIV-1 particles, highlighting Env clustering and the Env-specific labelling. Scale bar: 200 nm

observation spot over a line that spans several μm[15, 16]. Scanning FCS in general offers increased parallelisation (multiple correlation curves from a single measurement) and reduced photobleaching[17, 18].

Here, we describe sSTED-FCS measurements of molecular mobility on the surface of individual HIV-1 particles. We find that molecular dynamics of virus surface Env is two orders of magnitude slower than on the plasma membrane. Env mobility is dependent on maturation status as well as Gag–Env interactions via Env C-terminal tail (CT). Additional diffusion data of glycophosphatidylinositol (GPI)-anchored proteins as well as major hisocompatibility complex class-I (MHC-I) highlights a generally slow molecular mobility on the viral membrane surface, which is caused by its high degree of lipid order.

## Results

**HIV-1 surface Env mobility can be determined by sSTED-FCS.** To establish whether sSTED-FCS is capable of measuring Env mobility on the surface of individual virus particles, we used wild-type mature HIV-1 particles based on the well-described replication incompetent provirus construct pCHIV[19]. To identify the position of individual particles enhanced green fluorescent protein-tagged viral accessory protein Vpr (eGFP.Vpr) was incorporated into the virus using a co-transfected plasmid[20]. Env was visualised by immunolabeling with Fab fragments of 2G12 anti-gp120 domain antibody.

First, we tested whether immobilised wild-type HIV-1 particles can be visualised in live unfixed state and without the addition of microscopy embedding media that might affect virus surface dynamics. STED imaging of Env signal from Abberior STAR 635P-labelled 2G12 Fab immunocomplexes overlaid on confocal images of the eGFP-tagged viruses (Fig. 1b, c and Supplementary Fig. 1) revealed strong and specific staining with distinct Env patterns as seen in previous studies[2, 11].

To acquire sSTED-FCS data, individual virus particles were aligned with the trajectory of the scan line using eGFP.Vpr signal as a guide (Fig. 2a). Afterwards, the microscope was switched

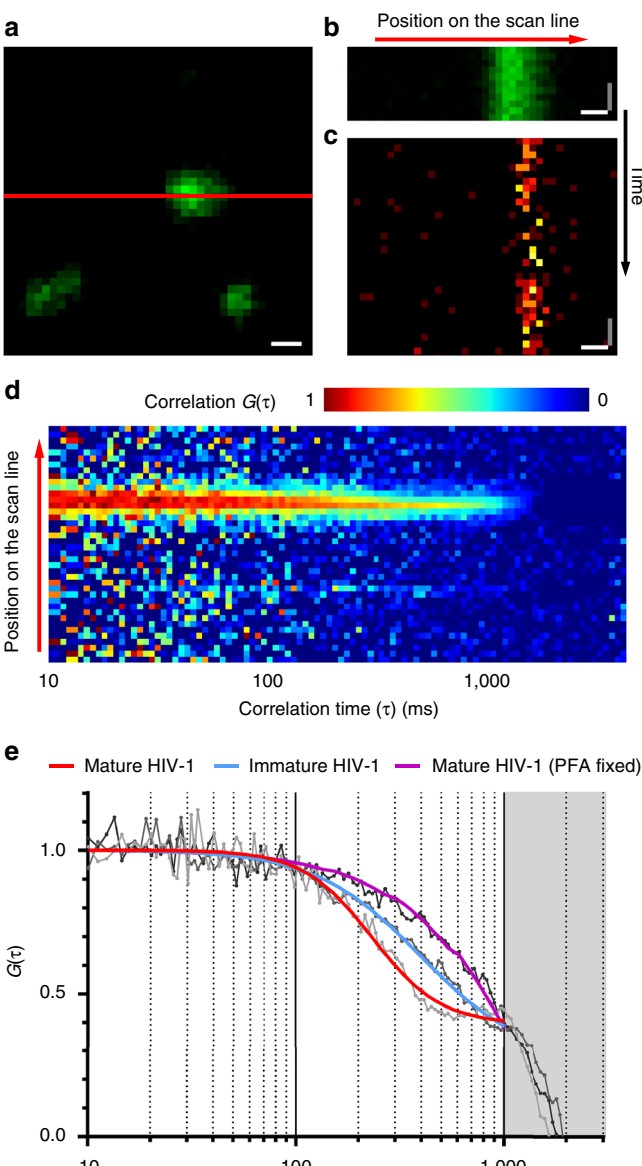

**Fig. 2** sSTED-FCS measurements of Env diffusion on HIV-1 particles. **a** Live confocal imaging was used to locate HIV-1 virus particles in a 2 μm × 2 μm imaging window using the eGFP.Vpr signal (*green*) as a guide and to align them with the position of the scan line (*red*). Scale bar: 200 nm. **b**, **c** Representative signal intensity carpets for eGFP.Vpr (*green*) in confocal mode and Env (*orange*) in STED mode on wild-type mature HIV-1 particles. Image x- and y-axis correspond to the position on the scan line and signal intensity at each time point, respectively. Scale bars: x-axis (*white*) = 200 nm, y-axis (*grey*) = 4.4 ms. **d** Correlation carpet generated from signal intensity carpet shown in **c**. Image x- and y-axis correspond to correlation time τ and the position on the scan line, respectively. Colour code corresponds to the normalised autocorrelation curves G(τ) at each position on the scan line. **e** Representative normalised autocorrelation curves of Env diffusion (*grey* and *black* lines) obtained from individual positions on the scan line within correlation carpets. Autocorrelation curves were fitted (*coloured lines*) for mature (*red*), immature (*blue*) and PFA fixed HIV-1 particles (*purple*) using generic two-dimensional diffusion model. *Greyed out area* corresponds to the photobleaching-only portion of the correlation data

from standard imaging (x–y-axis) to intensity carpet (x–t-axis) mode to record fluorescence intensity fluctuations at each position (or pixel) along the scanned line over time. First, a short intensity carpet was acquired for eGFP.Vpr signal in confocal mode to ensure that the position of the scanned line was correctly overlapped with the position of one or several virus particles and to record the pixel positions of the virus (Fig. 2b). Afterwards, an intensity carpet was acquired for Env signal in STED mode (Fig. 2c and Supplementary Fig. 2). Intensity carpets in both confocal and STED mode have shown no detectable drift during the acquisition times (Supplementary Fig. 2). Scanned fluorescence fluctuations were recorded with the following parameters: scanning frequency 0.9 kHz, scan line length 2 μm, pixel dwell time 20 μs, total measurement time 5 s, pixel size 50 nm/pixel and observation spot diameter 55 nm full-width-at-half-maximum (FWHM) (Supplementary Fig. 1).

Acquisition parameters were chosen to take into account (a) a small enough observation spot size that would allow for the successful acquisition of diffusion data from subdiffraction sized virus particles (Supplementary Fig. 3), (b) a line scan frequency that would detect dynamics equivalent to plasma membrane protein diffusing through the chosen size observation spot[15, 21] (see Methods), (c) a length of the scanned line to easily search for and select individual virus particles, and (d) a pixel dwell time that would result in an acceptable signal to noise ratio at a minimum possible excitation power (5 μW at the back aperture of the objective) to minimise photobleaching.

Following sSTED-FCS Env signal acquisition, intensity carpets were autocorrelated and autocorrelation curves were generated for each pixel position on the scanned line (Fig. 2d). Only pixels corresponding to the location of the virus particle yielded usable Env signal autocorrelation curves. These regions were selected and the corresponding FCS data fitted with a generic two-dimensional (2D) diffusion model (Fig. 2e) to obtain the average transit times of labelled Env molecules through the excitation spot at that location. FCS data with a sole exponential decay characteristic caused by excessive photobleaching[22] were discarded (Fig. 2e, *purple*) and only FCS data displaying sigmoid decay characteristics were used to establish average transit times for wild-type HIV-1 particles (Fig. 2e, *red*). Analysis of the FCS data obtained from individual mature HIV-1 particles revealed a median Env transition time $t_{xy}$ = 260 ms (Supplementary Fig. 4) through 55 nm excitation spot corresponding to a median diffusion coefficient $D$ = 0.002 μm²/s (Fig. 3a).

HIV-1 Env FCS data were found to feature a high degree of anomalous diffusion (with an anomaly factor $α > 1$, see Methods) indicating that the behaviour of the Env molecules on HIV-1 surface might not be completely modelled by the 2D diffusion equation. The high $α$-values were found to be independent of Env mobility (diffusion coefficient) or a correction factor that accounts for overall photobleaching (Supplementary Fig. 5), highlighting that values of $α > 1$ were not caused by photobleaching (see Methods). To test whether 2D diffusion equation is an adequate fitting model, we performed further analysis using a fitting model considering the diffusion on the surface of a spherical vesicle or the probability of photobleaching before the complete transit through the observation spot[22] (see Methods and Supplementary Methods). Fitting with these alternative models revealed similar results as the generic 2D diffusion equation (Supplementary Fig. 6). Therefore, despite high $α$-values, this fitting model was used in all subsequent experiments.

To ensure that the diffusion coefficients determined for Env were not biased by staining procedure artefacts such as the effect of 2G12 Fab fragment binding to specific gp120 epitope, we acquired Env mobility data using Fab fragment of another anti-gp120 antibody, b12 (Supplementary Fig. 7). Env mobility

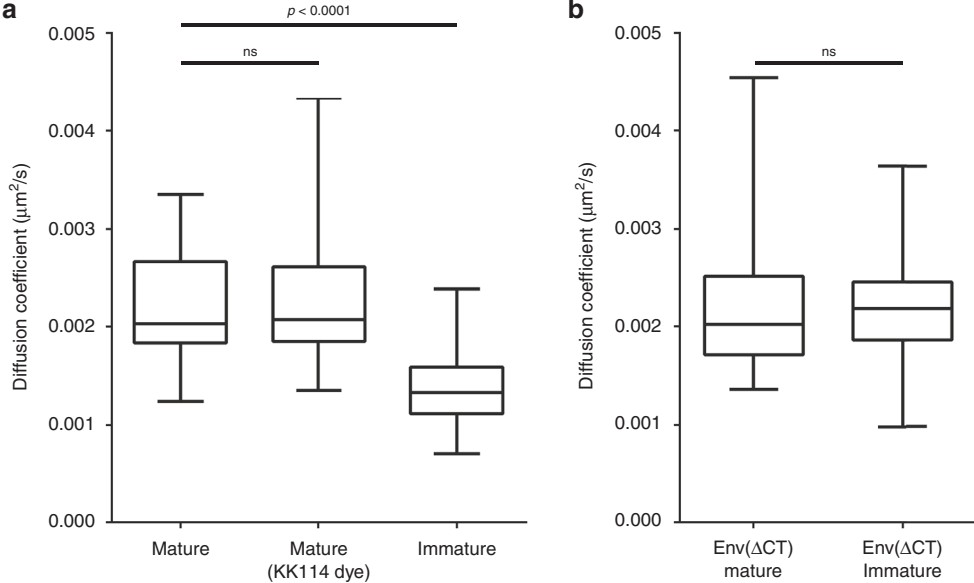

**Fig. 3** Diffusion coefficients of HIV-1 Env in mature, immature and Env C-terminal tail truncated particles. Median Env diffusion coefficients ($D$) were determined by sSTED-FCS measurements of 50 particles each from two independent virus preparations. Results are shown in a box (IQR) and whisker (min/max) plot and the statistical significance was assessed by Wilcoxon rank-sum test. (**a**) $D$-values of mature, immature and mature particles labelled with an alternative dye (KK114). (**b**) $D$-values of mature and immature particles pseudotyped with Env(ΔCT)

measurements obtained for this antibody ($D = 0.0023 \ \mu m^2/s$) were the same as for 2G12 suggesting that obtained Env mobilities were not affected by the choice of Env epitope. Furthermore, to exclude the effect photobleaching artefacts, we acquired sSTED-FCS Env mobility data using a structurally different far red dye for Env labelling. Measurements using the dye Abberior STAR RED (KK114) as label yielded very similar median Env diffusion coefficient ($D = 0.0021 \ \mu m^2/s$) as before (Fig. 3a, Supplementary Fig. 4), indicating that the observed Env mobility was independent of dye photophysics/chemistry. We have also recorded sSTED-FCS data of Env mobility after paraformaldehyde fixation. Measurements of wild-type HIV-1 particles pre-fixed with 3% paraformaldehyde solution for 30 min have abolished most of the recordable mobility as expected, displaying pure photobleaching (sole exponential decay of the FCS data) (Fig. 2e), with only a small subset of the observed particles displaying a very slow transit times ($t_{xy} \approx 400 \ ms$) (Supplementary Fig. 4) likely owing to the incomplete fixation.

**Env mobility depends on maturation and Gag–Env interactions.** Following the establishment of the technique to measure Env mobility on HIV-1 surface, we tested whether the virus maturation status influences Env mobility. Here, we compared diffusion coefficients of Env on wild-type mature particles with those on genetically immature pCHIV PR(−) virus particles. Measurements of Env mobility on the surface of immature virus particles yielded a significantly slower mobility than of mature wild-type, with transit times of $\approx 400 \ ms$, corresponding to a median diffusion coefficient $D \approx 0.0013 \ \mu m^2/s$ (Fig. 2e, *blue*, Fig. 3a, Supplementary Figs. 2, 4). These results are close to those obtained for paraformaldehyde (PFA)-fixed conditions, especially for those measurements where the FCS data were not dominated by an exponential decay owing to predominant photobleaching (Supplementary Fig. 4). This indicates that Env mobility changes significantly as HIV-1 undergoes maturation, with more mobile conditions in the mature and close to immobile conditions in the immature case.

To elucidate the mechanism behind the relationship between HIV-1 maturation and Env mobility we have investigated the role

of 151 amino-acid-long Env C-terminal tail (CT) that interacts with underlying MA domain of Gag[23]. Here we used mature and immature pCHIV HIV-1 particles pseudotyped with CT-truncated variant of Env and determined the mobility of this Env(ΔCT) variant. In both cases of mature and immature virus, Env(ΔCT) displayed median diffusion coefficients $D = 0.002 \ \mu m^2/s$ and $D = 0.0021 \ \mu m^2/s$, respectively, which are indistinguishable from those obtained for Env on mature wild-type HIV-1 (Fig. 3b and Supplementary Fig. 4). This suggests that although the absence of Env CT does not affect the overall Env mobility, the rigid underlying Gag lattice in the immature viruses appears to immobilise Env molecules via CT interactions until it is fully disassembled during maturation, thus allowing Env molecules to move more freely on the virus surface.

**HIV-1 envelope is intrinsically a low mobility environment.** Low Env mobility suggests that HIV-1 particle surface may be a very immobile environment. Hence, we investigated whether this is due to Env unique structure and interactions or generally low molecular dynamics of HIV-1 envelope. To accomplish this we compared virus surface and plasma membrane molecular dynamics of Env with other proteins that become incorporated into HIV-1 membrane during assembly:[24] 20 kDa SNAP tag protein fused to GPI anchor (GPI-SNAP) and 54 kDa MHC-I membrane protein.

For the plasma membrane mobility analysis, 293T cells were transfected with either Env or GPI-SNAP expression constructs, whereas for MHC-I analysis endogenously expressed cell protein was used. The expressed cell surface proteins Env, MHC-I and GPI-SNAP were stained with membrane impermeable Abberior STAR RED (KK114)-labelled anti-gp120 (2G12) Fab immunocomplex, anti-MHC-I (W6/32) Fab immunocomplex and SNAP tag substrate, respectively (Fig. 4a–c). Their diffusion properties were analysed using FCS and STED-FCS[14]. Our results have shown that Env exhibited a heterogeneous mobility in the plasma membrane with a median value of the diffusion coefficient $D = 0.15 \ \mu m^2/s$ (Fig. 4d). This was 100-fold faster than in the HIV-1 envelope (as outlined before) but also significantly slower than GPI-SNAP ($D = 0.34 \ \mu m^2/s$) or MHC-I ($D = 0.25 \ \mu m^2/s$) in the plasma

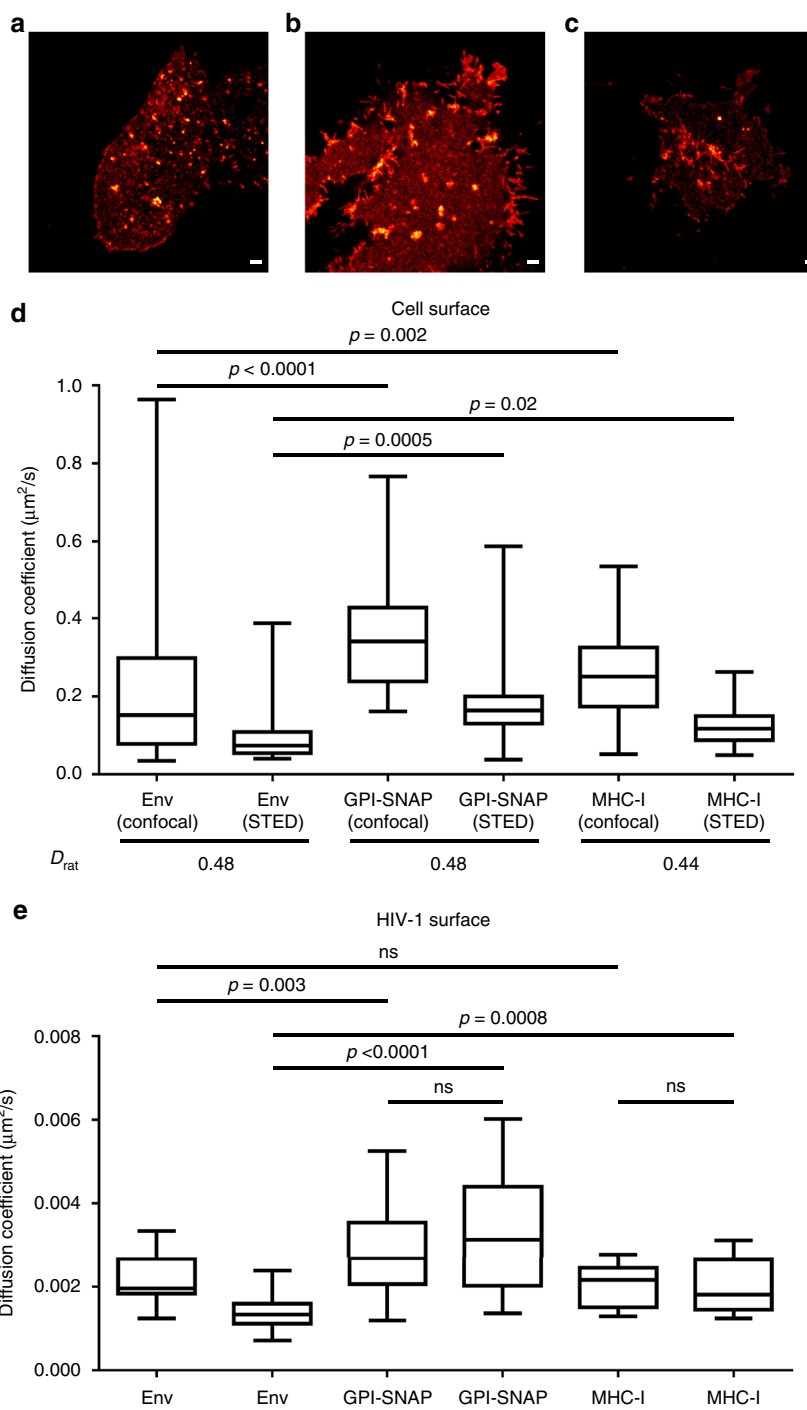

**Fig. 4** Comparison of protein molecular dynamics in cell and virus membranes. **a–c** Live confocal images of 293T cells expressing Env **a** and GPI-SNAP **b** supplied in trans, and endogenous MHC-I **c**. Env and MHC-I were visualised by immunostaining and GPI-SNAP by fluorescent SNAP tag substrate. Scale bars: 2 μm. **d** Median diffusion coefficients ($D$) of Env, GPI-SNAP and MHC-I cell surface mobility were determined by confocal FCS ($D_{conf}$, observation spot diameter = 240 nm) and STED-FCS ($D_{STED}$, observation spot diameter = 55 nm) measurements from 10 cells each in three independent experiments. Results are shown in a box (IQR) and whisker (min/max) plot. Ratios $D_{rat} = D_{STED}/D_{conf}$, given below the x-axis, indicate trapped diffusion characteristics for all proteins ($D_{rat} < 1$). The statistical significance was assessed by Wilcoxon rank-sum test. **e** $D$-values of Env, GPI-SNAP and MHC-I diffusion in mature and immature HIV-1 particles were determined by sSTED-FCS analysis of 50 particles each from two independent virus preparations. Results are shown in a box (IQR) and whisker (min/max) plot and the statistical significance was assessed by Wilcoxon rank-sum test

membrane. This difference in diffusion between Env and other proteins may be due to the comparative protein size, effects of Fab immunolabelling as well as the presence of interacting proteins.

Diffusion coefficients obtained from confocal ($D_{conf}$, observation spot 240 nm FWHM) and STED ($D_{STED}$, observation spot 55 nm FWHM) recordings revealed a ratio $D_{STED}/D_{conf}$ ($D_{rat}$) < 1 for all proteins in the plasma membrane (Fig. 4d), which indicates trapped diffusion, that is, millisecond-long arrests at points of interactions with more immobile structures or binding partners, as previously observed for sphingolipids in cell membranes[14].

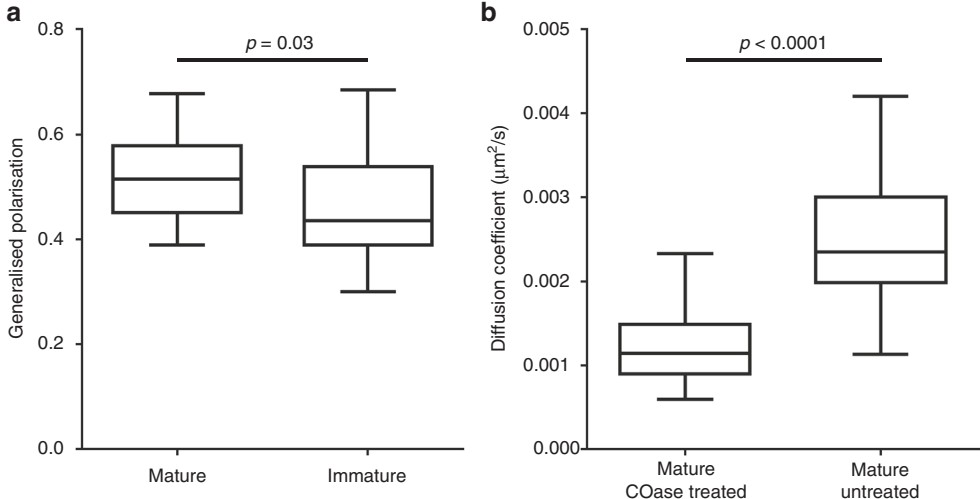

**Fig. 5** Analysis of HIV-1 lipid membrane properties by C-Laurdan staining and after cholesterol depletion. **a** Median values of the generalised polarisation (GP) parameters were determined from pixels of spectral images corresponding to C-Laurdan and mCherry.Vpr labelled mature and immature virus particles. Results represent measurements from 40 particles each from two independent virus preparations and are shown in a box (IQR) and whisker (min/max) plot. The statistical significance was assessed using Wilcoxon rank-sum test. **b** Median Env diffusion coefficient (D) of cholesterol oxidase (COase) treated HIV-1 was determined by sSTED-FCS measurements of 50 mature particles each from two independent virus preparations. Results are shown in a box (IQR) and whisker (min/max) plot and the statistical significance was assessed by Wilcoxon rank-sum test

Measurements of the mobility of GPI-SNAP and MHC-I on mature and immature HIV-1 surfaces revealed median diffusion coefficients $D_{\mathrm{mature}} = 0.0027\ \mu m^2/s$, $D_{\mathrm{immature}} = 0.0031\ \mu m^2/s$ for GPI-SNAP and $D_{\mathrm{mature}} = 0.0021\ \mu m^2/s$, $D_{\mathrm{immature}} = 0.0018\ \mu m^2/s$ for MHC-I (Fig. 4e). The mobility of GPI-SNAP on the virus was found to be significantly higher (1.35- to 1.5-fold) than the values of the diffusion coefficient of Env on the surface of mature HIV-1 (Fig. 4e; note that this factor is in the same range as the 2.3-fold difference of $D$-values for these proteins in the plasma membrane). On the other hand, MHC-I mobility on mature HIV-1 surface was similar to that of Env. Despite these unique characteristics, which may be related to the protein size, type of membrane anchoring and/or the degree of interaction with viral lipids and proteins, both proteins displayed the same general trend as Env where their mobility became 100-fold lower on HIV-1 surface compared to the plasma membrane. Moreover, unlike Env, HIV-1 surface mobility of GPI-SNAP and MHC-I was not affected by the virus maturation state (Fig. 4e).

We next investigated possible reasons for the ~100-fold reduced protein mobility in the HIV-1 envelope. A possible explanation could lie in an increased molecular crowding. Previous ensemble or model system measurements have, for example, highlighted the high degree of lipid order or packing in the envelope membrane[25, 26], mainly owing to the increased portion of saturated lipids and cholesterol compared to cellular plasma membranes[27, 28]. We refined these measurements by determination of lipid packing on individual mature and immature HIV-1 viruses using the polarity-sensitive dye C-Laurdan in combination with spectral scanning microscopy. In this technique spectral shifts in the fluorescence emission of C-Laurdan indicate changes in the lipid packing, quantified by values of the generalised polarisation (GP) parameter, which is negative for fluid, less-packed lipid environments and positive for rigid, highly packed ones[29]. GP values of individual C-Laurdan labelled HIV-1 particles ranged between 0.51 and 0.45, irrespective of the maturation state (Fig. 5a). These GP values are in line with previous bulk studies of mature particles and match those of highly ordered sphingomyelin-cholesterol model membranes[25, 26] or giant-plasma membrane vesicles[29]. This similarity indicates that HIV-1 envelope is in itself a very rigid, immobile

environment that may slow down the diffusion of integrated proteins. In addition, it suggests that lipid ordering is independent of the virus maturation status, and hence the observed changes in Env mobility are not induced by changes in the membrane organisation of mature and immature viruses.

To further test the importance of HIV-1 membrane organisation for Env mobility, we then sought to alter HIV-1 envelope characteristics by reducing its cholesterol content. Cholesterol is known to alter the fluidity of the membrane environment, for example, increasing the fluidity in highly ordered membranes. Previous studies have established HIV-1 envelope cholesterol as a critical component for virus infectivity[30]. These studies employed methyl-β-cyclodextrin for extracting cholesterol from the membrane envelope. However, this treatment can cause additional effects on the target membrane such as induction of solid-like regions[31]. In contrast, Cholesterol Oxidase (COase) treatment does not remove existing cholesterol molecules, and thus does not change the total number of molecules in HIV-1 membrane. This enzyme converts membrane-modulating cholesterol into cholestenone, leading to an alteration of membrane properties such as lipid packing. Env mobility measurements of COase-treated immobilised mature HIV-1 particles show a significant reduction of diffusion coefficient ($D \approx 0.0011\ \mu m^2/s$) (Fig. 5b), similar to the one observed in the largely immobile immature viruses (Fig. 3a). This observation confirms the highly ordered nature of the HIV-1 membrane envelope, as the reduction of cholesterol in such environments leads to reduced membrane fluidity and thus decreased mobility of embedded molecules.

## Discussion
Little is known about the dynamic properties of the HIV-1 virus surface. This study describes measurements and analysis of molecular mobility on the surface of individual HIV-1 particles and its relationship with virus maturation. Using sSTED-FCS approach we have established that, in wild-type mature HIV-1, Env has a measurable, but a very low surface mobility. Despite acquisition settings that would enable for the detection of mobilities equivalent to plasma membrane protein diffusion, we did not detect any faster moving molecules, indicating that low mobility may be a true characteristic of the viral membrane.

On the other hand, the sSTED-FCS approach is not without limitations. First, despite a high increase in resolution our study was still ultimately constrained by 55 nm sized observation spot. Although such spot was sufficient to detect Env mobility on subdiffraction sized viruses it is still just under the half of the size of the average virus ( < 140 nm). Therefore, it was still not enough to allow for the discrimination of mobility between different areas of the virus surface such as the centre or the edges and thus our results represent an average protein mobility over the entire virus surface. Second, the acquisition of the intensity carpet for Env was still affected by photobleaching, limiting the acquisition time to only several seconds (Supplementary Fig. 2) and making this technique unsuitable for the acquisition of long-term mobility changes performed on the same virus particle. However, as the photobleaching is more apparent owing to the low number of Env molecules, the acquisition time may be extended when investigating the diffusion characteristics of more abundant molecules on virus surfaces. Finally, the current fitting of FCS data may not accurately reflect the true mobility due to, for example, the highly curved nature of the virus or the photo-bleaching. Although we addressed this by employing different FCS fitting procedures, including those that correct for spherical vesicle diffusion and for photobleaching, the high degree of anomaly resulting from the simple 2D diffusion model indicate that the obtained diffusion coefficients may still only represent our best interpretation given the available tools, yet still accurately reflecting relative changes in mobility.

Despite these limitations, our study has established that there is a significant difference in the relative mobility between mature and immature particles. Low Env mobility in mature virus was found to be further reduced in immature virus particles, indicating that Env lateral diffusion is dependent on virus maturation status. This pattern of changes in dynamic properties of HIV-1 is in line with results of the study that demonstrated a link between HIV-1 particle stiffness in mature and immature virus and entry competence[7], highlighting that changes in surface properties of the virus are an important mechanism that enables for entry of mature virus only. Furthermore, our findings suggest an underlying mechanism for previously described Env clustering that is required for a productive virus entry[2]. Here, the disassembly of the rigid Gag lattice that holds Env immobile via MA-Env CT interactions enables Env to move on the virus surface and self-assemble into an entry competent cluster. Further studies with Env(ΔCT) variant indicated that Env CT appears to be essential for this process and its absence decouples Env from underlying Gag lattice giving it wild-type-like mobility that is independent of the virus maturation state. This in turn enables for Env re-clustering in immature Env(ΔCT) mutants upon contact with CD4 receptor and restoration of ability for virus entry. Such Env(ΔCT) mediated decoupling was also observed in studies of virus stiffness[7], further supporting an important role of Gag–Env interactions in changes of virus surface properties during maturation. Interestingly, Env CT truncation in mature virus did not appear to alter Env mobility compared with wild-type Env indicating that this region and its interactions with MA does not influence Env protein mobility in mature virus.

HIV-1 surface protein mobility was found to be two orders of magnitude slower than in the plasma membrane where diffusion coefficients of tested proteins were similar to that of many other proteins in the plasma membrane environment[32]. This trend was found to be true irrespective of whether the protein is the essential HIV-1 surface protein Env, externally supplied membrane GPI-anchored protein GPI-SNAP or host cell protein MHC-I. The general difference between plasma membrane and HIV-1 surface mobility and lack of dependence on virus maturation status for GPI-SNAP and MHC-I indicates that the observed very slow protein diffusion on the virus surface is not caused by direct interactions with underlying virus proteins but rather due to the properties of the viral membrane itself. Thus $D \approx 0.002\text{-}0.003 \ \mu m^2/s$ appears to represent a general intrinsic mobility that may be shared by all HIV-1 surface proteins present on the wild-type mature particle.

The very low-mobility environment of the HIV-1 surface may be explained by various aspects of virus morphology. For example, a passive incorporation of a large variety of cellular proteins during virus budding[24] as well as the tightly packed internal virus structure[33] might contribute to limiting the movement of the virus surface proteins. Moreover, in terms of the lipid environment, previous studies have shown that HIV-1 envelope composition is enriched in cholesterol and saturated lipids[27, 28]. This environment favours high lipid packing leading to a large molecular crowding that may lower surface protein mobility. Our C-Laurdan based analysis of lipid packing of individual HIV-1 particles confirmed that the virus envelope is a very rigid environment (GP values ≈ 0.5), that is independent of the maturation status. This suggests that observed changes in Env mobility are not induced by changes in the membrane lipid packing but rather appear to be caused by the interaction of Env with the underlying Gag lattice. Furthermore, as shown by COase treatments, the highly packed nature of HIV-1 envelope appears to be an important factor in low virus surface Env diffusion with further decrease in membrane fluidity leading to Env immobilisation. Previous studies have also shown that alterations to HIV-1 membrane lipid composition[28, 30] or protein crowding[34] lead to defects in virus entry/infectivity. Taken together, we speculate that alterations in HIV-1 membrane would lead to inability for Env to form entry competent clusters during maturation, lack of Env protein mobility that may be required during fusion and finally a failure of the altered virus envelope to fuse with the cell target membrane. Therefore, any therapies altering HIV-1 envelope packing would have a potential to target virus at multiple points during virus replication cycle.

Our study has provided insights into a very slow molecular mobility within subdiffraction sized highly crowded virus envelope. The dependence of Env mobility on virus maturation provides a new tool to study virus maturation kinetics without any alterations to the virus structural protein Gag. Env mobility measurements may also be used to identify maturation inhibitors as well as drugs that alter the properties of virus membranes and thus may provide avenues for new anti-HIV therapy. Moreover, our measurements provide a way for a determination of molecular mobility on different subdiffraction-sized and highly curved particles, such as other virus types or vesicles, allowing for the investigation of their dynamic properties.

## Methods

**Plasmids and cells**. Replication-incompetent HIV-1 particles were produced using pCHIV plasmid[19], expressing all HIV-1$_{NL4-3}$ proteins except Nef and lacking the viral long-terminal repeat sequences. pCHIV derivatives containing a mutation in the PR active site (pCHIV PR(−)) and/or premature Env termination (pCHIV Env(−)) were described previously[19, 35]. Pseudotyping was performed using plasmids pCAGGS expressing an Env variant with a 144 amino-acid truncation of its CT (Env(ΔCT))[36] or pGPI-SNAP. The cell plasma membrane expression experiments were performed using plasmids pCAGGS expressing wild-type Env (Env(wt)) or GPI-SNAP as well as endogenously expressed MHC-I. HIV-1 expression plasmid pCHIV and its derivatives as well as plasmids expressing Env(ΔCT), Env(wt) and mCherry.Vpr were provided by Barbara Müller and Hans-Georg Kräusslich. Plasmids expressing eGFP.Vpr[20] and GPI-SNAP were provided by Tom Hope and the NanoBiophotonics group, Max–Planck-Institute of Biophysical Chemistry, Gottingen, respectively.

293T cells (ATCC CRL-3216) were grown in Dulbecco's modified Eagle's medium (Sigma), supplemented with 10% fetal calf serum, 100 U/ml penicillin-streptomycin and 20 mM HEPES pH 7.4. Cells were maintained at 37 °C, 5% CO2.

**Antibodies and dyes**. Human anti-gp120 monoclonal antibody 2G12 and b12 were purchased from Polymun Scientific, and mouse anti-MHC-I monoclonal antibody W6/32 was produced in house from hybridoma stocks[37]. Fab fragments were generated using the Fab Micro Preparation kit (Pierce) according to the manufacturer's instructions. Anti-human and anti-mouse IgG Fab fragments (Jackson ImmunoResearch) were coupled to Abberior STAR 635P or Abberior STAR RED (KK114) dyes (Abberior GmbH, Göttingen, Germany) via NHS-ester chemistry according to the dye manufacturer's instructions.

**Virus particle preparation and purification**. Virus particles were prepared from the tissue culture supernatant of 293T cells co-transfected using polyethyleneimine with 14 μg pCHIV or pCHIV PR(−) and 1 μg peGFP.Vpr or pmCherry.Vpr per 10 cm dish, respectively. For pseudotyped particles, 15 μg of pCHIV Env(-), 1 μg of peGFP.Vpr and 2.5 μg of the indicated Env expression plasmid were used.

Tissue culture supernatants were harvested 48 h after transfection, cleared by filtration through a 0.45 μm nitrocellulose filter, and particles were purified by ultracentrifugation through 20% (w/w) sucrose cushion at 70,000 g (avg.) for 2 h at 4 °C. For C-Laurdan experiments, particles were further purified by ultracentrifugation through 6–35% OptiPrep (Sigma) gradient at 250,000 g (avg.) for 90 min at 4 °C in. Virus containing gradient fractions were diluted in phosphate-buffered saline (PBS) and pelleted at 70,000 g (avg.) for 2 h at 4 °C. Particles were resuspended in ice-cold 20 mM HEPES/PBS pH 7.4, snap frozen and stored in aliquots at −80 °C. All ultracentrifugation steps were performed in SW 41 Ti rotor.

**Cell transfection**. 293T cells were transfected with plasmids expressing wild type Env or GPI-SNAP using TurboFect transfection reagent (Thermo Fisher Scientific) according to manufacturer's specifications.

**Microscope setup**. Imaging and FCS based measurements were performed on a STED microscope based on a modified Abberior Instrument RESOLFT QUAD-P super-resolution microscope (Abberior Instruments GmbH) equipped with three pulsed excitation lasers (640, 594, and 485 nm; LDH-D-C-640P and LDH-D-C-485P; Picoquant, Berlin, Germany, and PDL 594; Abberior Instruments) with 80 ps pulse width and a pulsed STED laser (Titanium:Sapphire laser system, MaiTai; Newport Spectra-Physics Ltd, Didcot, UK) operating at 780 nm and 80 MHz repetition rate[38]. STED laser pulses were stretched to a pulse width of ~ 250–350 ps using two 20 cm SF6 optical glass rods and a 120 m-long single-mode polarisation maintaining optical fibre (OZ Optics Ltd, Lutterworth, UK). Shuttering and power adjustment of the STED laser were controlled with an acousto-optical modulator (MT110-B50A1.5-IRHK, AAA/Photon Lines Ltd, Banbury, UK). Control over the circular polarisation and generation of a donut shaped focal intensity distribution was achieved by incorporation of half-wave/quarter-wave plates (B. Halle, Berlin, Germany) and a phase-modifying vortex plate (VPP-1a, RPC Photonics, Rochester, NY), respectively, into the optical path of the STED laser beam. STED and excitation laser beams were spatially superimposed and the fluorescence light was separated using appropriate dichroic filters (ZT740SPRDC, AHF Analysentechnik, Tübingen, Germany). The temporal synchronisation of laser pulses was achieved by triggering the excitation lasers by the STED laser using a photodiode (APS-100-01, Becker & Hickl, Berlin, Germany). An on-board FPGA card (Abberior Instruments) was used for time alignment control between the laser pulses. Positioning and scanning of laser foci was realised using the QUAD beam scanner unit of the Abberior system for lateral directions, and an objective lens positioning system (MIPOS 100PL, Piezosystem Jena, Jena, Germany) for the axial direction. The fluorescence excitation and collection was performed using a 100 × /1.40 NA UPlanSApo oil immersion objective (Olympus Industrial, Southend-on-Sea, UK). The fluorescence signal was descanned, passed through an adjustable pinhole (Thorlabs Limited, Ely, UK) and detected by a single photon counting avalanche photo diode (SPCM-AQRH-13, Excelitas Tecologies) with appropriate fluorescence filters (AHF Analysentechnik). All acquisition operations were controlled by Imspector software (Abberior Instruments) and point FCS data were recorded using a hardware correlator (Flex02-08D, correlator.com, operated by the company's software).

C-Laurdan measurements were performed on a Zeiss LSM 780 confocal microscope equipped with a 32-channel GaAsP spectral imaging detector[29].

**Microscope sample preparation**. For calibration of the observation spot sizes using STED-FCS, a supported lipid bilayer (SLB) was prepared by spin coating a coverslip with a solution of 1 mg/ml 1,2-Dioleoyl-sn-glycero-3-phosphocholine (Avanti Polar Lipids, Alabaster, AL) and 0.5 μg/ml Abberior STAR RED (KK114)-DPPE (1,2-dihexadecanoyl-sn-glycero-3-phosphoethanolamine; Abberior GmbH) in CHCl₃/MeOH[39]. Coverslips were pre-cleaned and etched by piranha solution (3:1 sulfuric acid and hydrogen peroxide). The lipid bilayer was formed by rehydration with SLB buffer containing 10 mM HEPES and 150 mM NaCl pH 7.4.

For Env mobility measurements, purified virus particles were adhered to poly-L-lysine (Sigma) coated glass coverslips for 30 min. Coverslips were blocked using 2% bovine serum albumin (BSA) (Sigma)/PBS for 15 min. Particles were stained for Env using 10 ng/μl 2G12 or b12 anti-gp120 Fab fragments and anti-human Abberior STAR 635P or Abberior STAR RED (KK114) conjugated Fab fragments

for 1 h each in blocking buffer. Immunostained particles were washed and mounted in PBS, followed by sSTED-FCS analysis. All steps were carried out at room temperature.

For C-Laurdan measurements, Optiprep purified virus particles expressing mCherry.Vpr were incubated with 20 μM C-Laurdan for 20 min at room temperature. Unbound C-Laurdan was removed by ultracentrifugation through 20% (w/w) sucrose cushion at 70,000 g (avg.) for 2 h at 4 °C. Purified C-Laurdan labelled HIV-1 was adhered to poly-L-lysine coated coverslips for 15 min and washed with PBS.

Env, GPI-SNAP and MHC-I expressing cells were labelled at 16 °C to minimise endocytosis of the label. Samples were blocked using 0.5% BSA/Leibovitz's L-15 Medium (L-15) for 15 min. Env-expressing cells were immunostained using 10 ng/μl 2G12 anti-gp120 Fab fragments and anti-human Abberior STAR RED (KK114) conjugated Fab fragments for 1 h each in the blocking buffer. GPI-SNAP expressing cells were labelled using 3.6 μM BG-Abberior STAR RED (KK114) conjugate for 2 h in the blocking buffer. Endogenous MHC-I was visualised using 10 ng/μl W6/32 anti-MHC-I Fab fragments and anti-human Abberior STAR RED (KK114) conjugated Fab fragments for 1 h each in the blocking buffer. Following labelling, cells were washed and overlaid with L-15, followed by FCS and STED-FCS analysis.

**Cholesterol oxidase treatment**. Coverslip adhered and Env immunostained virus particles were incubated with 0.5 U COase from *Streptomyces* sp. (Sigma) for 30 min at 37 °C. Treated particles were washed and mounted in PBS, followed by sSTED-FCS analysis.

**sSTED-FCS and (STED)-FCS signal acquisition**. sSTED-FCS data of labelled Env on HIV-1 surface were acquired at room temperature using eGFP.Vpr signal as a virus position guide. Env signal intensity carpets were recorded using the Imspector software with the following parameters: scanning frequency 0.9 kHz, scan line length 2 μm, pixel dwell time 20 μs, total measurement time 5 s, pixel size 50 nm/pixel, 5 μW excitation power (back aperture) at 640 nm, observation spot diameter 55 nm FWHM (as determined by SLB calibration measurements). The line frequency $F$ was set according to the requirement of being much faster than the expected average transit times $t_{xy}$ of the investigated molecules through the observation spot, $F > 4/t_{xy}$[15]. Assuming a diffusion coefficient of $D = 0.12$ μm²/s for membrane-embedded proteins, we estimated an average transit time $t_{xy} = d^2/(8\ln(2)D) \approx 5$ ms through the 60 nm large observation spot (diameter $d = 60$ nm full-width-at-half-maximum, FWHM), that is, we set the line scan frequency to at least $F = 0.8$ kHz[15].

Point (STED)-FCS data of labelled Env, MHC-I and GPI-SNAP on the surface of 293T cells was recorded at room temperature at the following parameters: total measurement time 15 s, 5 μW excitation power (back aperture) at 640 nm, observation spot diameter 240 nm FWHM for confocal and 55 nm FWHM for STED recordings, and 80 MHz laser repetition rate.

**Photobleaching estimation**. To estimate the degree of photobleaching in the acquired sSTED-FCS intensity carpets, a photobleaching correction factor (PBC, $1/t_b$) was calculated. First, intensity carpets were spatially cropped in $x$-axis to contain only those pixels corresponding to the virus position. The decay of the fluorescence signal intensity $f(t)$ over time $t$ from those regions was then fitted with an exponential function:[40]

$$f(t) = f_0 \exp(-t/t_b) \tag{1}$$

where $f_0$ denotes the initial fluorescence signal at $t = 0$ and $t_b$ the average time it takes to photobleach by a factor of 1/e.

**FCS curve autocorrelation and fitting**. The custom-designed (Python based) FoCuS-scan software was used to autocorrelate the scanning FCS data, and to fit them with a generic 2D diffusion model including an anomalous factor ($\alpha$) to allow for non-ideal diffusion:

$$G_N(\tau) = C + G_N(0)\left[1 + \left(\tau/t_{xy}\right)^{\alpha}\right]^{-1} \tag{2}$$

where $G_N(\tau)$ is the correlation function at time lag $\tau$, $C$ the offset, $G_N(0)$ the amplitude, $t_{xy}$ the average lateral transit time through the observation spot, and $\alpha$ is an anomaly factor which takes into account possible deviations (for example, owing to photobleaching or large curvature) from the assumed purely lateral 2D diffusion ($\alpha = 1$ ideal case).

We also tested a model aiming to correct for photobleaching:

$$G_N(\tau) = C + G_N(0)\left[1 + \left(\tau/t_{xy}\right)\right]^{-1}[1 - B_a + B_a \exp(-K_z\tau)] \tag{3}$$

where $k_z$ is the rate constant for photobleaching (as determined from the exponentially decaying FCS data in PFA fixed Env) and $B_a$ is an amplitude accounting for the relative contribution of photobleaching to the FCS data (free floating)[22]. Here, we do not make use of an anomalous factor instead attempting to correct any inconsistencies from the 2D diffusion model by considering the photobleaching.

Owing to tight spherical/curved structure of the HIV-1, outer envelope we also tested a fitting model that takes into account the diffusion on the surface of a spherical/highly curved vesicle:

$$G_N(\tau) = C + G_N(0) \left( \sum_{l=0}^{\infty} (2l+1) \left[ \int_{-1}^{1} P_l(x) \exp\left(\frac{a^2 x^2}{2\sigma^2}\right) \right]^2 e^{-(D/a^2)l(l+1)\tau} \right) \quad (4)$$

where $a$ represents the vesicle radius, $x$ a substituted integral prefactor, $\sigma$ the variance of the Gaussian-assumed observation spot, $D$ the diffusion coefficient, and $P_l$ the Legendre polynomial with coefficient l. A full theoretical explanation of this equation is available in the Supplementary Methods section.

The values resulting from the fits were tabulated for each fitted curve. Values of the molecular diffusion coefficient ($D$) were calculated from $t_{xy}$:

$$D = (FWHM \times 2\ln(2))^2 / (4t_{xy}) \quad (5)$$

where $FWHM$ represents the usual full-width half maximum of the observation spot.

**C-Laurdan imaging and generalised polarisation quantitation**. C-Laurdan measurements were acquired by spectral imaging as described previously[29]. C-Laurdan was excited at 405 nm and the emission spectrum was collected between 415 nm and 549 nm in 8.9 nm wide intervals. Generalised polarisation (GP) values were determined for regions with positive mCherry.Vpr (virus) signal:

$$GP = \frac{I_{L_o} - I_{L_d}}{I_{L_o} + I_{L_d}} \quad (6)$$

where $I_{L_o}$ and $I_{L_d}$ are the fluorescence intensities at 510 nm and 440 nm, respectively, as extracted from the recorded fluorescence spectrum[29].

**Statistical analysis**. Owing to the Poisson-like distribution of the diffusion data, Wilcoxon test, which does not assume Gaussian distribution, was used with $P < 0.05$ considered statistically significant. Tests were performed using Graphpad Prism software. Power calculations confirmed that for chosen sample sizes the power of a two-sided hypothesis test at the 0.05 significance was over 90%. In all compared groups interquartile range (IQR) serves as an estimate of variation. IQRs were similar in all compared groups with the exception of virus particles observed with 110 nm and 240 nm observation spots (Supplementary Fig. 3) as well as PFA fixed virus (Supplementary Fig. 4). These control data sets correspond to invalid detection and immobile conditions, respectively. They are predominated by photobleaching only correlation curves which result in apparent > 400 ms transition times when fitted.

**Data availability**. The data that support the findings of this study are available from the corresponding authors upon request.

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

## Acknowledgements

We would like to thank Barbara Müller and Hans-Georg Kräusslich (University of Heidelberg, Germany) for the Env(wt), Env(ΔCT), mCherry.Vpr, pCHIV based plasmids and fruitful discussions as well as Thomas J. Hope (Northwestern University, Chicago) and the Department of NanoBiophotonics (Max Planck Institute for Biophysical Chemistry, Gottingen, Germany) for the eGFP.Vpr and GPI-SNAP constructs, respectively. We also grateful to Dilip Shrestha (Weatherall Institute of Molecular Medicine, Oxford, UK) for producing W6/32 Fab fragments. J.C. and C.E. are supported by the MRC (grant number MC_UU_12010/unit programs G0902418 and MC_UU_12025), MRC/BBSRC/EPSRC (grant MR/K01577X/1), Wellcome Trust (grant 104924/14/Z/14 and Strategic Award 091911 (Micron)), Deutsche Forschungsgemeinschaft (Research unit 1905 "Structure and function of the peroxisomal translocon"), and Oxford internal funds (EPA Cephalosporin Fund and John Fell Fund). D.W. is supported by funding from the MRC and EPSRC (grant number EP/L016052/1). P.C. is supported by a predoctoral grant from the Basque Government and N.H. by the NIH (grant number AI097051). J.E. is grateful for financial support by the Deutsche Forschungsgemeinschaft (SFB 937, project A07).

## Author contributions

J.C. and C.E. conceived the study. J.C. carried out all the experiments except the determination of virus GP values by C-Laurdan spectral imaging microscopy performed by P.C. with assistance from N.H. D.W. wrote the scanning FCS data autocorrelation and fitting software. S.G. supported building and optimising the STED-FCS microscope. J.E. developed a mathematical proof for the diffusion on the surface of a spherical vesicle. J.C. and C.E. wrote the manuscript. All authors contributed in discussing the data, experiments and the manuscript.

## Additional information

**Competing interests:** The authors declare no competing financial interests.

