## [Peer Review File · Nature Communications]

Reviewers' comments:

Reviewer #1 (Remarks to the Author):

In this study, the authors use scanning fluorescence correlation spectroscopy (sFCS) on a super-resolution STED microscope to examine the mobility of the HIV-1 envelope (Env) glycoprotein on immature vs. mature virions. They observe that Env mobility increased upon particle maturation, and that removal of the long Env (gp41) cytoplasmic tail (CT) reverses this effect. Diffusion of both Env and a GPI-anchored test protein was markedly slower on the HIV-1 particle than on the plasma membrane. The data support previous findings that the HIV-1 lipid bilayer is substantially more rigid than the plasma membrane, and suggest that interactions between the gp41 CT and uncleaved Gag limit Env mobility in the immature particle.

In general, the data are of broad interest and the results provide some fresh insights into Env behavior on the HIV-1 viral particle. A few issues, however, need to be addressed.

1. For non-microscopy experts, some of the data will be difficult to follow. For example, the uninitiated reader will have no idea what is being shown in Fig. 2d, and this is not adequately explained. Likewise, supp Fig. 2 will not be clear to many readers.
2. Some of the conclusions reached in this paper were anticipated from AFM studies from the Rouso lab. This work should be cited and discussed.
3. The authors state several times that the slower mobility of full-length Env on immature vs. mature particles is a result of interactions between Gag and the gp41 CT. While this is likely to be the case, the authors provide no evidence, and alternative interpretations are possible. This point should be toned down.
4. Minor, "vpr" is misspelled on p. 11, line 280 and gp120 is misspelled on line 288.

Reviewer #2 (Remarks to the Author):

The lead author (Chojnacki) has previously used STED microscopy to show that HIV envelope proteins undergo clustering redistribution on the membrane surface during virion maturation (PMID: 23112332). In this manuscript, Chojnacki has worked with new mentor Eggeling to continue this line of investigation by incorporating scanning Fluorescence Correlation Spectroscopy (FCS) within the context of STED. More specifically, they interrogate whether the HIV Env proteins are able to fluctuate differently within its local environment before and after virion maturation.

I understand that the authors have used mobility to describe their observation, but this assessor feels that 'fluctuation' might be a more appropriate description with the data, as this assessor feels that the word mobility is slightly too strong of a description.

Authors have used mature HIV, immature HIV (by genetic inactivation of HIV protease), and HIV Env cytoplasmic tail deletion mutant in their analysis. Authors have also used laurdan

dye based GP ratio in an attempt to understand the lipid packing of the viral envelope.

Overall, this assessor feels that the combination of FCS with STED is an interesting and exciting possibility, and should be supported.

At the same time, this assessor also feels that the current dataset mark the beginning of an excellent story, but there are a number of issues must be addressed prior to publication. Some of these suggestions (although require additional and significant amount of new experiments, but they should be within the reach of the authors). At the moment, it feels a lot like phenomenon description without clearly explain the importance of these fluctuations (motilities) are.

This assessor also feels that a number of interpretations made by the authors have unintentionally overstated their findings and they should be addressed prior to publication.

This assessor is not an expert in FCS, and has taken the calculations and measurements of FCS on face values. This assessor would recommend editor to have this aspect being looked at independently by other researchers in the field.

Specific Comments are listed below:

1. This assessor feels the measurement is perhaps best described as fluctuation rate rather than diffusion. In order to make the claim of diffusion works, authors should perform additional measurement with other virion associated membrane proteins as control to get the trends of biological properties of these membrane proteins. I would imagine a couple more proteins that are known to be rafts and non-raft-associated would be very helpful;
2. Also, the authors will need to use alternative labelling procedure to light up Env to ensure that the fluctuation (mobility) rate is not a consequence (or at least bias) by binding with broadly neutralising antibody;
3. Authors should show how Env fluctuation (mobility) rate change upon CD4 binding and/or CD4 and co-receptor engagement;
4. My understanding of FCS would be to show the fluctuation of fluorophores within the so call sampling focusing area. If I read the manuscript correctly, the spot diameter of this work is 55nm FWHM. There is no question STED-FCS is a great achievement in itself. However, given the HIV particle would be 120nm in diameter to begin with, this spot diameter will cover more than $\frac{1}{4}$ or $\frac{1}{3}$ of the bird's eye view surface area. Therefore, one would wonder whether 50nm spot diameter is small enough? In particular, would 'not seeing an effect' really mean no changes? Consequently, authors should interpret their data even more careful than they are now by toning down their claims accordingly;
5. Somewhat a similar question / logic with the Laurdan dye GP ratio. It is unfortunate the Laurdan dye GP ratio was not done in the STED mode, if so, the data would be much more meaningful. Current values appear to be average of events, and this assessor feels that the authors have over-interpreted their data, as well as missed a great opportunity to reveal important biological properties of HIV virion membrane;
6. Using Laurdan approach only to evaluate lipid contributions in HIV membrane domains on Env fluctuation / mobility is limited and underdone. Authors should apply their STED-FCS

approach on sphingolipid deficient viruses (PMID: 16481622), Cholesterol depleted and replaced HIV (PMID: 12857902), Phospholipid floppase HIV mutant (PMID 18241333). At least some of these lipid modified viruses should be included in the study if the authors would like to make claim on lipid packing and Env fluctuation (mobility) – in particular given non STED Laurdan info has limited value;

7. There is a logical problem of using a replication incompetent virus to obtain information of Env behaviour in real HIV. I understand the concern of biosafety. However, the authors should at least use a genome-less lentiviral vector that has pseudotyped HIV envelope for a small amount of confirmatory analysis to ensure such observed behaviour is not a consequence of pCHIV. Using the lentiviral system, authors will also be able to artificially increase the amount of HIV Env on virion particle surface. On one hand, it might make data collection a lot quicker, at the same time, authors will also be able to evaluate whether amount of Env on virion surface could interfere with Env fluctuation and mobility;

8. There are large number of cellular proteins found on the surface of virion particles. Many of them are known to rafts associated and some are not. It would be important to measure the mobility of some of these proteins as their corresponding internal controls

9. Implications of involvement of Env CT is interesting, and it fits with the current understanding of Env and Gag (and more specifically MA) relationship. Recent work of the Freed lab has used mutagenesis to reveal the linkage of MA trimerization, Env packaging, and how compensatory mutations influence the process (see PMID 24244165, 26711999), it is perhaps important for the author to provide additional info of HIV biology and the 'fluctuation' of Env signals using some of these MA mutants;

10. Information of GPI-SNAP fluctuation is useful. However, it would be important to show the rate of other virion-associated protein to reveal whether some sort of trend exists, or all proteins appear to have somewhat a different rate / number. It is important to find some of consents and explain what these data mean.

11. The structure of HIV Env will be different before it binds to its receptor and/or broadly neutralising Ab (such as 2G12. Would the authors be able to use another type of immuno tag (or different epitope mediated antibody tag) to show whether the same kind of behavior persist? i.e. observation is not biased by 2G12 or neutralising Ab.

12. Utilization of soluble CD4 receptor might help to reveal the potential biological significance of these fluctuation of Env on virion surface.

13. Will the authors be able to comment whether virion release itself is a determinant of the fluctuation of Env proteins?

Minor comments:

1. Even Env mobility can be used as a marker, these values are not easily extracted from experimental system – defeating the definition and purpose of marker;

2. Referencing of authors seems to be overly biased toward his current and former mentoring labs, a much more balanced referencing practice would be needed to reflect the contributions of the field;

3. It is puzzling that the authors did not use the photo-destructible PI in their system to synchronised the maturation process – such approach will truly reveal the dynamics. HIV maturation are very fast process, and I would imagine several second should be plenty. Have the authors done this experiment?

4. On page 3 line 40, is disassembly the right description? the CA lattice actually is 'maintained' and to proceed to become cone shape core, while both the arrangement of MA before and after the maturation are still not known!
- 5.

Reviewer #3 (Remarks to the Author):

The manuscript "Envelope glycoprotein mobility on HIV-1 particles depends on the virus maturation state" by Chojnacki et al. describes the results of a study taking advantage of the highly advanced technique of sSTED-FCS (scanning Stimulated Emission Depletion Fluorescence Correlation Spectroscopy). sSTED-FCS, first described in a Nature Communications publication in 2014 by the same group, is unique in its capabilities to measure dynamics on the ms-time scale in areas as small as about 60 nm diameter. Applying sSTED-FCS, the authors measure the diffusion coefficients of the Env protein in the membrane of HIV-1. They find that its mobility increases during the virus maturation process which can be explained through the interaction of Env with the Gag protein in the virus core. The authors show that Env as well as GPI-anchored proteins have much lower mobility (1 to 2 orders of magnitude lower diffusion coefficients) on the HIV-1 envelope than in cell membranes demonstrating that the HIV-1 envelope is a low mobility environment. Generalized polarization measurements using C-Laurdan suggest that the high lipid order of the envelope is the reason for this.

Generally, the manuscript is well written and easy to follow. A few typos, missing commas or small grammar errors which reduce the readability of a few sentences can be easily be corrected by an additional round of proof-reading.

Diffusion measurements on single virus particles are extremely challenging due to their small size well below the diffraction limit. STED-FCS is therefore ideally suited for these measurement. To my understanding, the presented work represents the first application of FCS to membrane systems that small, which is a remarkable achievement in itself! While I have a few technical concerns (see below), which I would like the authors to consider and discuss, I believe that the observation of a dramatically different diffusion behavior in the HIV-1 envelope compared to the cell membrane is an important finding and worthy of publication in Nature Communications.

Concerns:

- 1) Diffusion model (page 6, discussion, methods, and supplement): the authors briefly discuss which diffusion model should be used and emphasize that a simple 2D diffusion model cannot be correct. They test different models including one correcting for spherical vesicle diffusion, however, they do not end up applying it to their biological studies. It is puzzling why not and currently not well justified.
- 2) The diffusion behavior should look dramatically different between an observation spot centered on the virus particle and a spot located at the side of the virus. In the first case, the membranes (top and bottom of virus) are oriented more or less parallel to the focal

plane, in the second case, the membrane is oriented approximately parallel to the optic axis. How stringent did the authors pick the pixels used in their FCS analysis? To my understanding, only focus spots centered on the virus particles should be analyzed and not the neighboring pixels which represent a very different membrane orientation. I might have missed it but could not find any discussion of this selection/problem. Did the authors select only central pixels? If not, why not? I would naively expect that one should be able to measure different apparent diffusion coefficients for pixels representing the different membrane orientations. Was this observable?

Minor comment:

- Page 9, second paragraph: the authors claim that the observed bleaching makes sSTED-FCS unsuitable for the acquisition of long term mobility changes. But could that not be solved by scanning the same particle less frequently to stretch out the measurement over a longer time?

Reviewer #1:

In general, the data are of broad interest and the results provide some fresh insights into Env behavior on the HIV-1 viral particle.

We thank the reviewer for this positive conclusion.

Specific comments:

1. For non-microscopy experts, some of the data will be difficult to follow. For example, the uninitiated reader will have no idea what is being shown in Fig. 2d, and this is not adequately explained. Likewise, supp Fig. 2 will not be clear to many readers.

We thank the reviewer for pointing this out. We have revised the manuscript figures and figure legends (especially **Fig. 2** and **Supplementary Fig. 2**) to provide clearer explanations for non-microscopy experts.

2. Some of the conclusions reached in this paper were anticipated from AFM studies from the Rousso lab. This work should be cited and discussed.

We thank the reviewer for highlighting this and apologise for the omission. We have revised the manuscript to cite these important studies and discuss their findings in light of our results (Discussion section).

3. The authors state several times that the slower mobility of full-length Env on immature vs. mature particles is a result of interactions between Gag and the gp41 CT. While this is likely to be the case, the authors provide no evidence, and alternative interpretations are possible. This point should be toned down.

While we agree that alternative interpretations are possible, we believe that our experiments that show both maturation dependence on the mobility Env(Δ CT) mutant and lack of such dependence in another virus envelope associated membrane protein MHC-I provide a compelling evidence that this may be the case. These findings are also further reinforced by similar findings from Rousso's lab (PMID: 17158573) where Env(Δ CT) mutant also resulted in changes of virus surface properties. Nonetheless, we have revised the appropriate sections of the manuscript to tone this point down.

4. Minor, "vpr" is misspelled on p. 11, line 280 and gp120 is misspelled on line 288.

This mistake was corrected in the revised manuscript.

Reviewer #2:

Overall, this assessor feels that the combination of FCS with STED is an interesting and exciting possibility, and should be supported.

We thank the reviewer for this positive conclusion.

At the same time, this assessor also feels that the current dataset mark the beginning of an excellent story, but there are a number of issues must be addressed prior to publication. Some of these suggestions (although require additional and significant amount of new experiments, but they should be within the reach of the authors). At the moment, it feels a lot like phenomenon description without clearly explain the importance of these fluctuations (motilities) are.

Please note, our study has not only described the mobility of the surface molecules for a first time but also through the use of HIV mutants established that Env mobility may act as an underlying mechanism that allows for the Env molecules to form clusters only in mature virus particles thus enabling it to become entry competent. Establishment of mobility of Env molecules as an important aspect of HIV maturation and entry competence highlights it as a potential exploitation point for future HIV therapies.

I understand that the authors have used mobility to describe their observation, but this assessor feels that 'fluctuation' might be a more appropriate description with the data, as this assessor feels that the word mobility is slightly too strong of a description.

The nomenclature we use in this study is in line with that used in the field of fluorescence-based mobility studies (such as using fluorescence correlation spectroscopy or single-molecule tracking). Here, the words such as mobility and diffusion signify the degree of motion of molecules on the membrane surface or in a solution. Since our results have shown that that despite being very slow the diffusion coefficient of Env on virus surface is reproducibly measurable implying that these molecules move on the virus surface and hence there is no good reason not to refer to this motion as mobility. On the other hand, the word "(signal) fluctuations" commonly refers to the detected rise and fall in the fluorescence signal as molecules come in and out of the observation spot during FCS measurements. These fluctuations are then self-correlated to generate correlations curves. Therefore the use of the word fluctuations to refer to mobility would introduce unnecessary confusion for the readers.

This assessor also feels that a number of interpretations made by the authors have unintentionally overstated their findings and they should be addressed prior to publication.

Despite a vague nature of this comment, we believe it may be related to the comment 3 raised by Reviewer #1. As previously explained, additional experiments performed on MHC-I molecule provide additional evidence supporting our interpretations. We also revised the manuscript and toned down the language of our interpretations.

Specific comments:

1. This assessor feels the measurement is perhaps best described as fluctuation rate rather than diffusion. In order to make the claim of diffusion works, authors should perform additional measurement with other virion associated membrane proteins as control to get the trends of biological properties of these membrane proteins. I would imagine a couple more proteins that are known to be rafts and non-raft-associated would be very helpful;

8. There are large number of cellular proteins found on the surface of virion particles. Many of them are known to rafts associated and some are not. It would be important to measure the mobility of some of these proteins as their corresponding internal controls.

10. Information of GPI-SNAP fluctuation is useful. However, it would be important to show the rate of other virion-associated protein to reveal whether some sort of trend exists, or all proteins appear to have somewhat a different rate / number. It is important to find some of consents and explain what these data mean.

We thank the reviewer for these suggestions, as indicated by points 1, 8 and 10. To address it we performed additional experiments measuring a cell and virus surface mobility of MHC Class I, a host cell protein known to be incorporated into HIV particles (**Fig. 4**). Hence, our study now compares cell and virus surface mobilities of **a)** endogenous HIV membrane protein (Env), **b)** external GPI-anchored protein supplied *in trans* (GPI-SNAP) and **c)** endogenous host cell membrane protein incorporated into HIV particles (MHC-I). Data obtained from all these proteins suggests that, despite small differences in individual mobilities, the general trend is that all proteins on virus surface have very much reduced mobility compared to cell surface (Figure 4).

2. Also, the authors will need to use alternative labelling procedure to light up Env to ensure that the fluctuation (mobility) rate is not a consequence (or at least bias) by binding with broadly neutralising antibody;

11. The structure of HIV Env will be different before it binds to its receptor and/or broadly neutralising Ab (such as 2G12. Would the authors be able to use another type of immuno tag (or different epitope mediated antibody tag) to show whether the same kind of behavior persist? i.e. observation is not biased by 2G12 or neutralising Ab.

We thank the reviewer for raising this point. To address this potential issue, we performed an additional control experiment for Env mobility experiment using Fab fragments of another anti-Env antibody (b12) (**Supplementary Figure 8**). This antibody binds to a different epitope than 2G12. Results have shown than mobility obtained from measurements using this tag are similar to those from 2G12, hence results based on 2G12 tag do not appear to be biased by the choice of the antibody.

3. Authors should show how Env fluctuation (mobility) rate change upon CD4 binding and/or CD4 and co-receptor engagement;

12. Utilization of soluble CD4 receptor might help to reveal the potential biological significance of these fluctuation of Env on virion surface.

This is a useful suggestion and fully agree that measurements of Env mobility and its potential changes during receptor and co-receptor engagement would be useful to further the understanding of virus structure as well as its entry and fusion process. However, we feel that such investigations would fall outside of the scope of this study where we focussed on the dynamic properties of viruses alone. These experiments would require a development from scratch of a new experiment system where the virus attachment is synchronised with the target cell prior to mobility measurements. In addition, the measurement of virus Env mobility while in contact with the cell raises novel technical challenges and thus we feel that it would best constitute a separate project. Nonetheless, our future plans do include the adaptation of sSTED-FCS to measurements of molecular mobility during virus-cell interactions.

Regarding, the suggestion to use soluble CD4, unfortunately this compound is known to induce gp120 shedding (Thali, M. 1992, PMID: 1501286). Our experiments have shown that soluble CD4 treated virus particles loose majority of 2G12 signal indicating extensive gp120 shedding. Moreover, since soluble CD4 induced gp120 shed Env does not constitute normal Env behaviour we did not pursue mobility measurements of these molecules since we felt that obtained results would not add any accurate and therefore useful data for this study.

4. My understanding of FCS would be to show the fluctuation of fluorophores within the so call sampling focusing area. If I read the manuscript correctly, the spot diameter of this work is 55nm FWHM. There is no question STED-FCS is a great achievement in itself. However, given the HIV particle would be 120nm in diameter to begin with, this spot diameter will cover more than $\frac{1}{4}$ or $\frac{1}{3}$ of the bird's eye view surface area. Therefore, one would wonder whether 50nm spot diameter is small enough? In particular, would 'not seeing an effect' really mean no changes? Consequently, authors should interpret their data even more careful than they are now by toning down their claims accordingly;

We fully understand the reviewer's concern whether, given the subdiffraction size of the virus, the observation spot is small enough to observe Env molecules diffusing in and out of the surveillance area to provide valid FCS data. Therefore we have performed an experiment where we compared the data obtained from acquisitions at 55 nm, 110 nm and 240 nm (confocal) sized observations spots (**Supplementary Fig. 7**). Results have shown that only at 55 nm valid correlation curves could be acquired with photobleaching dominating the data for larger observation spots. This highlights that the 55 nm large observation spot is sufficient for observation of overall mobility on virus surface.

Furthermore, we are puzzled by a posed question about 'not seeing an effect'. In this study, we have clearly shown that our experimental conditions are sufficient to detect the mobility and its changes in the experiments testing the central themes of this study ie. a) are there any changes in Env mobility due to the virus maturation status, b) what are the differences in mobility between proteins in cell and virus membranes, and c) what are the effects of membrane alterations such as

PFA fixation or cholesterol oxidase treatment on Env mobility. As to the theoretical possibility of our settings not being sensitive to detect all the changes in the Env mobility, this question puts us in a difficult situation since it can be put towards any novel experimental system in existence. Since our investigation examines previously unstudied aspects of HIV, we cannot compare them with previous findings and fully understand that they represent our best interpretation given the technology. However, we are currently investigating alternative microscopy techniques that will enable for virus surface Env mobility measurements and thus provide the possibility to verify our findings in the future.

5. Somewhat a similar question / logic with the Laurdan dye GP ratio. It is unfortunate the Laurdan dye GP ratio was not done in the STED mode, if so, the data would be much more meaningful. Current values appear to be average of events, and this assessor feels that the authors have over-interpreted their data, as well as missed a great opportunity to reveal important biological properties of HIV virion membrane;

It has been shown multiple times that GP measurements represent meaningful and valuable results for quantifying lipid membrane packing. Here, GP measurements show that rigid lipid packing is independent of the virus maturation, and therefore it may be one of the causes behind the generally low protein diffusion on virus surface. Since unlike FCS, GP measurements do not rely on the diffusion in and out of the observation area, therefore they do not have a resolution requirement for study of subdiffraction sized objects. Hence they represent valid measurements of overall lipid packing in the virus particle.

We are also currently in the process of early evaluation of STED-based GP measurements (bioRxiv: 107334) and may use it in future projects to see if there are lipid packing differences within different regions of the virus e.g. ones with or without Env clusters. Unfortunately, STED-based GP measurements are not yet developed enough to be currently included as accurate read-out for this study; we will consider it for future experiments.

6. Using Laurdan approach only to evaluate lipid contributions in HIV membrane domains on Env fluctuation / mobility is limited and underdone. Authors should apply their STED-FCS approach on sphingolipid deficient viruses (PMID: 16481622), Cholesterol depleted and replaced HIV (PMID: 12857902), Phospholipid floppase HIV mutant (PMID 18241333). At least some of these lipid modified viruses should be included in the study if the authors would like to make claim on lipid packing and Env fluctuation (mobility) – in particular given non STED Laurdan info has limited value;

We thank the reviewer for this suggestion. To specifically investigate whether lipid packing affects surface protein mobility, we measured Env diffusion in Cholesterol Oxidase treated particles (**Fig. 5b**). Although we are grateful for the studies suggested by the reviewer as examples, harsh treatment like total cholesterol removal by methyl- β -cyclodextrin, can have a serious consequences on overall lipid membrane behaviour (Nishimura, SY. 2006, PMID: 16272447). Therefore we opted for a gentler Cholesterol Oxidase treatment that converts cholesterol into a non-active form (cholestenone) but does not remove it from the membrane. The results of this experiment shown that following the treatment Env mobility was lost, thus confirming that lipid packing is a critical determinant of Env mobility and thus virus infectivity.

7. There is a logical problem of using a replication incompetent virus to obtain information of Env behaviour in real HIV. I understand the concern of biosafety. However, the authors should at least use a genome-less lentiviral vector that has pseudotyped HIV envelope for a small amount of confirmatory analysis to ensure such observed behaviour is not a consequence of pCHIV. Using the lentiviral system, authors will also be able to artificially increase the amount of HIV Env on virus particle surface. On one hand, it might make data collection a lot quicker, at the same time, authors will also be able to evaluate whether amount of Env on virus surface could interfere with Env fluctuation and mobility;

We understand reviewer's concern but unfortunately, due to biosafety concerns, it is not possible to conduct FCS measurements on fully infectious virus (although we are currently in the early stages of establishing appropriate equipment in the biosafety category 3 environment for future studies of fully infectious pathogens). However, pCHIV which differs from full HIV only by a lack of transcription factor Nef and Long Terminal Repeat sequences, is a well-established model system that is structurally indistinguishable from the fully infectious virus and has been used in numerous studies probing the details of HIV assembly and maturation (PMID: 17097708, 19893629, 19553682, 19666477, 21394086, 23468635, 23115284, 24789789 and 27517329). In terms of Env it has been shown to incorporate the same amount as fully infectious virus (Chojnacki, J. 2012, PMID: 23112332) and therefore we consider it a highly suitable model system for study of dynamic properties of HIV. Hence we feel that the establishment and use of less structurally relevant genome-less model system would not be justified here.

However, although outside the scope of this study, we recognise that varying the amount of Env may bring additional insights into the mobility of proteins on virus surface especially since even related retroviruses incorporate larger amount of Env than HIV. We will therefore endeavour to investigate the importance of this phenomenon in our future studies.

9. Implications of involvement of Env CT is interesting, and it fits with the current understanding of Env and Gag (and more specifically MA) relationship. Recent work of the Freed lab has used mutagenesis to reveal the linkage of MA trimerization, Env packaging, and how compensatory mutations influence the process (see PMID 24244165, 26711999), it is perhaps important for the author to provide additional info of HIV biology and the 'fluctuation' of Env signals using some of these MA mutants;

Above studies have used mutations in MA domain of Gag in order to derive information about the structure of MA in HIV-1 particles and the nature of MA-Env interactions. To achieve this authors used combinations of MA mutants that either block or restore Env incorporation. Since our study deals with Env mobility, any mutations that block Env incorporation on virus surface cannot be analysed in our experimental system. Moreover, while our studies of Env C-terminal tail deleted mutant (Env(Δ CT)) have established that the deletion of this domain rescues the mobility of Env in immature particles, we observed no mobility differences between Env(Δ CT) and wild-type Env in mature particles. Since, as evidenced by a later part of our study, lipid packaging but not MA-Env interactions play a role in Env mobility, measuring it would not be useful in the study of nuances of MA-Env interactions.

13. Will the authors be able to comment whether virion release itself is a determinant of the fluctuation of Env proteins?

Unfortunately, we cannot comment on the importance of virus release since that would require establishment of a completely new cell based platform that includes a method of synchronising virus release prior to Env mobility measurements. Establishment of such platform would bring a host of new technical challenges and we therefore would consider it beyond the scope of this study that focused on virus particles alone.

Minor comments:

1. Even Env mobility can be used as a marker, these values are not easily extracted from experimental system – defeating the definition and purpose of marker

We are puzzled by this comment since although during establishment of our experimental system we have identified its limitations, it can still be used in HIV studies as a potential maturation marker, especially since it has unique benefit of not interfering with any of the virus internal structures. The difficulties in obtaining the data (which will be addressed as we refine our experimental setup for future studies) do not invalidate Env mobility itself as a maturation marker.

2. Referencing of authors seems to be overly biased toward his current and former mentoring labs, a much more balanced referencing practice would be needed to reflect the contributions of the field;

To address this point, we have revised our manuscript to include more comprehensive referencing of publications that underpin this study as well as studies that are important for a correct interpretation of our results.

3. It is puzzling that the authors did not use the photo-destructible PI in their system to synchronised the maturation process – such approach will truly reveal the dynamics. HIV maturation are very fast process, and I would imagine several second should be plenty. Have the authors done this experiment?

The estimated length of HIV-1 maturation is approx. 30 min long, as demonstrated by recent study (Hanne, J. 2016, PMID: 27517329). Therefore, while we performed preliminary tests of the photo-destructible PI system, it is currently not feasible to perform real time observation of the maturation dynamics on this time scale with our experimental setup. While we anticipate such experiments, their realization would require further optimization of the experimental approach, which is beyond the scope of the present study.

4. On page 3 line 40, is disassembly the right description? the CA lattice actually is 'maintained' and to proceed to become cone shape core, while both the arrangement of MA before and after the maturation are still not known!

Although, there are currently conflicting reports whether during the formation of a conical capsid in mature virus, CA lattice undergoes complete disassembly (Keller, PW. 2013, PMID: 24109217) or partial disassembly accompanied by rolling into a cone (Frank, GA. 2015, PMID: 25569620), the

word “disassembly” remains a valid term used in the literature, especially when used in the context of entire rigid immature Gag lattice rather than specifically CA domain of Gag.

Reviewer #3:

Diffusion measurements on single virus particles are extremely challenging due to their small size well below the diffraction limit. STED-FCS is therefore ideally suited for these measurements. To my understanding, the presented work represents the first application of FCS to membrane systems that small, which is a remarkable achievement in itself! While I have a few technical concerns (see below), which I would like the authors to consider and discuss, I believe that the observation of a dramatically different diffusion behavior in the HIV-1 envelope compared to the cell membrane is an important finding and worthy of publication in Nature Communications.

We thank the reviewer for this highly supportive assessment.

Specific comments:

1. Diffusion model (page 6, discussion, methods, and supplement): the authors briefly discuss which diffusion model should be used and emphasize that a simple 2D diffusion model cannot be correct. They test different models including one correcting for spherical vesicle diffusion, however, they do not end up applying it to their biological studies. It is puzzling why not and currently not well justified.

We apologise for this confusion. In our study, we noted that the use of the simple 2D diffusion mode resulted in uncharacteristically high anomaly factor (α) thus suggesting that this model may not be best suited for virus surface mobility analysis. Therefore, in addition to this simple model, we used photobleaching corrected and spherical vesicle diffusion models. Since fitting of those models gave us very similar results to the simple 2D model used initially (**Supplementary Fig. 6**), we decided to remain using it for all the subsequent analyses of virus surface protein mobility.

We have added an appropriate statement in the revised manuscript to clarify this.

2. The diffusion behavior should look dramatically different between an observation spot centered on the virus particle and a spot located at the side of the virus. In the first case, the membranes (top and bottom of virus) are oriented more or less parallel to the focal plane, in the second case, the membrane is oriented approximately parallel to the optic axis. How stringent did the authors pick the pixels used in their FCS analysis? To my understanding, only focus spots centered on the virus particles should be analyzed and not the neighboring pixels which represent a very different membrane orientation. I might have missed it but could not find any discussion of this selection/problem. Did the authors select only central pixels? If not, why not? I would naively expect that one should be able to measure different apparent diffusion coefficients for pixels representing the different membrane orientations. Was this observable?

We agree with the reviewer that this would be expected. However, during acquisition of correlation curves for Env mobility (whose pixel positions were chosen from the area co-aligning with eGFP.Vpr signal) we did not observe any correlation between what we considered the edge and centre positions. We believe that this may be due to a still limited 55 nm resolution which was used in this study. While sufficient to detect Env mobility on subdiffraction sized viruses

(Supplementary Fig. 7) it is still just under the half of the size of the average virus (< 140 nm). Therefore it may not be enough for us to discriminate between the virus centre and edges and our measurements represent an average mobility combining transitions in and out of the observation volume for both virus edge and the centre. We envisage that with continual improvement to the microscopes as well as the dyes used we will be able to further increase the resolution of our measurements to be able to discriminate which part of the virus they come from.

Minor comment:

- Page 9, second paragraph: the authors claim that the observed bleaching makes sSTED-FCS unsuitable for the acquisition of long term mobility changes. But could that not be solved by scanning the same particle less frequently to stretch out the measurement over a longer time?

Unfortunately, this proposed solution is not viable since relationship between scanning frequency and the degree of photobleaching is not linear. Therefore dropping the scan frequency does not result in any significant extension to the observation times. Moreover, high scanning frequency is a crucial parameter, even for the acquisition of the slow moving molecules, since it allows for a complete capture and subsequent correct fitting of the correlation curves.

REVIEWERS' COMMENTS:

Reviewer #1 (Remarks to the Author):

The authors have addressed my concerns.

Reviewer #2 (Remarks to the Author):

I feel the authors have mostly adequately addressed the concerns brought up from previous assessment. I am happy to support this manuscript for publication.

A couple minor points that I would like to respond to the authors.

- Some of those justifications and explanations in the rebuttal are very good for a non-expert such as myself, and thanks;

- the 'not seeing an effect' comment in the original assessment was meant for GPI-SNAP data, and now GPI-SNAP and MHC-I data in the revised manuscript. That is to say - would it be true that the GPI-SNAP and MHC-I are also 'mobile', but just below the detection limit? I certainly agree that the HIV Env mobility changes upon virus maturation, which is significantly different from either GPI-SNAP or MHC-I. Whether mobility of GPI-SNAP or MHC-I changes upon virus maturation would be difficult to comment if the movement range is within the 55nm range;

- Personally, I am still rather uncomfortable with the last statement of the abstract, which is

Our results establish HIV-1 Env mobility as a potential maturation MARKER and provide novel insights into dynamic properties of proteins on virus particle surfaces.

First, let me be clear that having the ability to measure HIV Env mobility in a virion particle is an excellent scientific achievement. However, I really cannot see 'Env mobility' can readily be used by others as a marker to measure virus maturation. This is because: (1) not that many people in the world would have the capacity to measure Env mobility in a single virus particle, therefore, such methodology will not be suitable for most researchers in the world; and (2) there are much easier way to measure virus maturation with simpler techniques, hence difficult for me to see 'measuring Env mobility' will gain wide stream acceptance as routinely utilised 'marker' to determine whether a virus has undergone maturation or not.

I also wonder whether it will be better to flip the last statement in the following way (or something similar) to emphasis the strength of the authors' findings - given the authors have already shown the mobility of Env changes during maturation.

Suggested new statement:

Our results on changing of Env mobility during maturation implies the local environment of

virion lipid membrane has undergone re-organization during maturation. These data provide novel insights into dynamic properties of proteins on virus particle surface

Happy to declare these are my personal interpretations of the data, and happy to go with whatever the authors, editor and other reviewers decide at the end.

Reviewer #3 (Remarks to the Author):

All my comments have been addressed satisfactorily. I recommend publication of the manuscript. However, I suggest to add a discussion along the lines of the authors' reply to my comment 2 also to the manuscript (either in the Discussion section or in the Supplement) since it provides important insights into the current limitations of the method.

Reviewer #1:

The authors have addressed my concerns.

We thank the reviewer for supporting the publication of our revised manuscript.

Reviewer #2:

I feel the authors have mostly adequately addressed the concerns brought up from previous assessment. I am happy to support this manuscript for publication.

We thank the reviewer for supporting the publication of our revised manuscript.

A couple minor points that I would like to respond to the authors.

-Some of those justifications and explanations in the rebuttal are very good for a non-expert such as myself, and thanks;

We are glad that our explanations were helpful. They will be available online for other researchers as well.

-the 'not seeing an effect' comment in the original assessment was meant for GPI-SNAP data, and now GPI-SNAP and MHC-I data in the revised manuscript. That is to say - would it be true that the GPI-SNAP and MHC-1 are also 'mobile', but just below the detection limit? I certainly agree that the HIV Env mobility changes upon virus maturation, which is significantly different from either GPI-SNAP or MHC-1. Whether mobility of GPI-SNAP or MHC-I changes upon virus maturation would be difficult to comment if the movement range is within the 55nm range;

Actually, our study has clearly shown that GPI-SNAP and MHC-I are still mobile (relative to Env) when incorporated in HIV-1 (**Fig. 4**) irrespective of virus maturation status. However, as also indicated by Reviewer #3, we acknowledge that, despite a high increase in resolution, our study was still ultimately limited by 55 nm resolution limit. While it was sufficient to detect Env mobility on subdiffraction sized viruses it is still just under the half of the size of the average virus (< 140 nm). Therefore, it might not be enough to discriminate between the virus centre and edges and thus our results represent an average protein mobility over the entire virus surface. Hence, there is a possibility that proteins may behave slightly differently depending on the position on the virus surface. This may be the case especially in immature particle, whose Gag shell is incomplete and underlies approx. 70% of the virus envelope. We have indicated this point in the Discussion section.

-Personally, I am still rather uncomfortable with the last statement of the abstract, which is

Our results establish HIV-1 Env mobility as a potential maturation MARKER and provide novel insights into dynamic properties of proteins on virus particle surfaces.

First, let me be clear that having the ability to measure HIV Env mobility in a virion particle is an excellent scientific achievement. However, I really cannot see 'Env mobility' can readily be used by others as a marker to measure virus maturation. This is because: (1) not that many people in the world would have the capacity to measure Env mobility in a single virus particle, therefore, such methodology will not be suitable for most researchers in the world; and (2) there are much easier way to measure virus maturation with simpler techniques, hence difficult for me to see 'measuring Env mobility' will gain wide stream acceptance as routinely utilised 'marker' to determine whether a virus has undergone maturation or not.

I also wonder whether it will be better to flip the last statement in the following way (or something similar) to emphasis the strength of the authors' findings - given the authors have already shown the mobility of Env changes during maturation.

Suggested new statement:

Our results on changing of Env mobility during maturation implies the local environment of virion lipid membrane has undergone re-organization during maturation. These data provide novel insights into dynamic properties of proteins on virus particle surface

Happy to declare these are my personal interpretations of the data, and happy to go with whatever the authors, editor and other reviewers decide at the end.

We thank the reviewer for this suggestion and the justification. We have amended the abstract to remove the statement on Env as a potential maturation marker.

Reviewer #3:

All my comments have been addressed satisfactorily. I recommend publication of the manuscript. However, I suggest to add a discussion along the lines of the authors' reply to my comment 2 also to the manuscript (either in the Discussion section or in the Supplement) since it provides important insights into the current limitations of the method.

We thank the reviewer for supporting the publication of our manuscript. We have added the appropriate point into the Discussion section.